# Assessing the Effects of Different Fillers and Moisture on Asphalt Mixtures' Mechanical Properties and Performance

Yongcai Liang [1], Tao Bai [1], Xiaolong Zhou [1,*], Fan Wu [1], Changlong Chenxin [1], Chao Peng [2,*], Luis Fuentes [3], Lubinda F. Walubita [4], Wei Li [5] and Xingchen Wang [5]

[1] School of Civil Engineering and Architecture, Wuhan Institute of Technology, Wuhan 430073, China
[2] Faculty of Engineering, China University of Geosciences, Wuhan 430074, China
[3] Civil and Environmental Engineering Department, Universidad del Norte, Barranquilla 25260, Colombia
[4] Texas A&M Transportation Institute (TTI), The Texas A&M University System, College Station, TX 77840, USA
[5] Shandong Hi-Speed Maintenance Group Co., Ltd., Jinan 250032, China
* Correspondence: zhouxiaolong@wit.edu.cn (X.Z.); pengchao@cug.edu.cn (C.P.)

**Abstract:** This laboratory study was conducted to comparatively assess the effects of different fillers and moisture on the mechanical properties and performance of asphalt mixtures. In the study, a typical Pen70 base asphalt was modified with four different filler materials, namely limestone powder, cement, slaked (hydrated) lime, and brake pad powder, to produce different asphalt mortars that were subsequently used to prepare the asphalt mixtures. Thereafter, various laboratory tests, namely dynamic uniaxial repeated compressive loading, freeze-thaw splitting, and semicircular bending (SCB) were conducted to evaluate the moisture sensitivity, high-temperature stability, low-temperature cracking, and fatigue performance of the asphalt mixtures before and after being subjected to water saturation conditions. Overall, the study results indicated superior moisture tolerance, water damage resistance, and performance for slaked (hydrated) lime, consecutively followed by brake pad powder, cement, and limestone powder. That is, for the materials evaluated and the laboratory test conditions considered, limestone mineral powder was found to be the most moisture-sensitive filler material, whilst slaked (hydrated) lime was the most moisture-tolerant and water-damage resistant filler material.

**Keywords:** asphalt; cement; brake pads; limestone; lime; asphalt mixture; moisture sensitivity; water damage; bonding; interface; mechanical properties

## 1. Introduction

Water damage in asphalt roads largely occurs through the intrusion of water into the pavement structure resulting in the strength degradation of the asphalt mixture. Under the action of traffic loading, the pavement gradually endures distresses such as potholing, spalling, striping, and structural damage [1]. In asphalt roads, water damage usually occurs through two mechanisms, namely cohesive and/or adhesive damage. Cohesive damage refers to the bonding deterioration within the asphalt itself under the action of water intrusion, causing the asphalt binder between the aggregates to crack and/or debond [2]. Adhesive damage, on the other hand, refers to the moisture progressively invading into the interface between the asphalt and aggregate or simply defined as the progressive deterioration that happens when water degrades the adhesive bond between the aggregate surface and the asphalt. This makes the asphalt membrane gradually peel off from the surface of the aggregates, leading to a loss in the cohesive force between the aggregates [3]. One method to improve the water stability of an asphalt mixture (or asphalt concrete) is aggregate gradation modifications or the use of additives such as fillers to either: (a) reduce moisture intrusion into the mixture, or (b) improve the adhesion between the asphalt and aggregate surfaces. Filler accounts for more than 80% of the total surface area of the mineral aggregates and can interact with the asphalt on its surface to give the resultant

asphalt matrix (or asphalt mortar) strong cementing abilities. That is, the filler not only contributes to the cementing and stability properties, but it also plays a filling role in the asphalt mixture. Therefore, the choice of a suitable filler to improve the water stability of the asphalt mixture is very critical.

At present, the most used filler material for road construction in China is limestone powder. However, cement, slaked lime, and waste materials are also often used as fillers for road construction in China [4,5]. In their study, Guo et al. [6,7] demonstrated that limestone powder can potentially increase the adhesion characteristics of the asphalt mortar, with a corresponding improvement in the overall performance of the asphalt mixture and the resultant pavement structure. Wang and Tian [8,9], through laboratory experimentations, found that the use of cement filler can significantly improve the water stability of asphalt mixtures. The networked hydration products resulting from the cement related reactions have been reported to be responsible for increasing the viscosity of the asphalt mortar, improving the interfacial bonding between with the aggregate surfaces and enhancing the mechanical properties of asphalt mixture.

On the other hand, Al-Tameemi and Grajales et al., [10,11] found that slaked lime can react with some specific functional groups in the asphalt and form a strong waterproofing compound that significantly improves the water stability of the asphalt mixture. Likewise, Hu et al., [12–15] successfully used waste brake pads as fillers to modify the asphalt mixture and found satisfactory results with respect to enhancing both the moisture tolerance and high-temperature stability performance of the asphalt mixture.

As reported in the literature, the effects of traditional fillers on the water stability of asphalt mixtures are often evaluated in the laboratory using freeze-thaw splitting tests [16]. Other laboratory tests, such as the Hamburg Wheel Tracking Tester (HWTT), have also been used successfully to characterize the moisture sensitivity, water damage, and striping potential of asphalt mixtures with different filler materials in both dry and wet conditions, i.e., before and after water saturation [17–19]. In the field, non-destructive technologies such as the Ground Penetrating Radar (GPR) have been indirectly used as indicative measures of the subsurface moisture damage in asphalt mixtures [20].

In general, most of the literature reviewed suggests conducting multiple laboratory tests to adequately characterize moisture damage in both asphalt mastics and asphalt mixtures, with the surface energy concepts exhibiting more promising potential [21,22]. However, as discussed above, the majority of the literature agrees that adding minerals fillers, including industrial by-products and nanomaterials, have the potential to improve the moisture tolerance, mechanical properties, performance characteristics, and durability of both asphalt mastics and asphalt mixtures [23–27]. Furthermore, the use of these fillers, particularly industrial by-product wastes, have the inherent benefit of contributing to environmental conservation and sustainability [28–30].

With the above background, this laboratory study was conducted to comprehensively evaluate the mechanical properties of asphalt mixtures modified with different filler materials when subjected to moisture conditioning. The primary goal of the study was to quantitatively study the effects of different fillers on the performance of the asphalt mixture before and after water saturation. To achieve this objective, the laboratory tests performed included dynamic uniaxial repeated compressive loading, freeze-thaw splitting, and semi-circular bending (SCB) for moisture sensitivity, high-temperature stability, low-temperature cracking, and fatigue evaluation of the asphalt mixtures before and after being subjected to water saturation conditions. As documented herein, four filler materials, namely limestone mineral powder, P·O42.5 cement, slaked (hydrated) lime, and waste brake pad powder, were comparatively evaluated in this study.

## 2. Materials and Test Methods

### 2.1. Raw Materials

2.1.1. Aggregates

Limestone with good hard texture, low flaky, and good angularity was used for both the coarse and fine aggregates. The filler, namely limestone mineral powder, comprised of finely grinded limestone ore. As shown in Tables 1 and 2, the physical properties of both the coarse and fine aggregates were in line with the Chinese standard requirements based on specification JTG E42-2005 [31].

**Table 1.** Physical and technical properties of the coarse aggregates.

| Test Item | | Test Results | Specification Requirement |
|---|---|---|---|
| Water absorption (%) | | 0.3 | $\leq 2.0$ |
| Crash value (%) | | 21.8 | $\leq 28$ |
| Needle and flake particle content (%) | | 9.8 | $\leq 15$ |
| Apparent specific gravity | 0–5 mm | 2.811 | |
| | 5–10 mm | 2.875 | $\geq 2.50$ |
| | 10–15 mm | 2.834 | |
| Los Angeles wear value (%) | | 15.8 | $\leq 28$ |

**Table 2.** Physical and technical properties of fine aggregates.

| Test Item | Test Results | Specification Requirement |
|---|---|---|
| Apparent specific gravity | 2.705 | $\geq 2.5$ |
| Angularity (flow time method) (s) | 36 | $\geq 30$ |
| Sand equivalent (%) | 76.8 | $\geq 60$ |

2.1.2. Filler Materials

Four filler materials, namely limestone ore powder (denoted as mineral powder), waste brake pad powder (denoted simply as brake pads), P·O42.5 cement (simply denoted as cement), and slaked lime were evaluated in the study. For consistency, all the fillers comprised of the finely grinded materials passing on the 0.075 mm sieve. Each of the four filler materials is described in the subsequent text.

Limestone mineral powder: The mineral powder was sourced from a limestone ore powder quarry in Hubei, China. Its key chemical composition and physical properties are shown in Tables 3 and 4, respectively. As listed in Table 4, the measured physical properties of the mineral powder were consistent with the Chinese specification JTGE42-2005 [31].

**Table 3.** Chemical composition of limestone.

| Element | $SiO_2$ | $CaO$ | $MgO$ | $Al_2O_3$ | $Fe_2O_3$ | Others |
|---|---|---|---|---|---|---|
| Proportion (%) | 1.8 | 52.8 | 1.4 | 1.8 | 0.4 | 41.8 |

**Table 4.** Physical properties of the limestone mineral powder.

| Test Item | Test Results | Specification Requirement |
|---|---|---|
| Apparent specific gravity | 2.639 | / |
| Particle size range (mm) | <0.075 | / |
| Specific surface area ($m^2$/g) | $7.8404 \pm 0.0108$ | / |
| Exterior | No lumps | / |

P·O42.5 cement: The cement used was P·O42.5 cement. Its chemical composition and physical properties are presented in Tables 5 and 6, respectively.

**Table 5.** Chemical composition of the P·O42.5 cement.

| Element | SiO₂ | CaO | MgO | Al₂O₃ | Fe₂O₃ | SiO₃ | K₂O | Na₂O | Others |
|---|---|---|---|---|---|---|---|---|---|
| Proportion (%) | 30.3 | 41.1 | 1.5 | 12.5 | 3.4 | 2.9 | 1.0 | 0.6 | 9.9 |

**Table 6.** Physical properties of Cement.

| Test Item | Test Results | Specification Requirement |
|---|---|---|
| Apparent specific gravity | 3.072 | / |
| Specific surface area (m²/g) | 2.0303 ± 0.0327 | / |
| Exterior | No caking granules | / |

Slaked (hydrated) lime: The chemical composition of the slaked lime used in this study is shown in Table 7, whilst the physical properties are summarized in Table 8. Note that in this paper that the term "slaked lime" has been used interchangeably to mean "hydrated lime".

**Table 7.** Chemical composition of slaked lime.

| Element | Ca (OH)₂ | Magnesium And Alkali Metals | Acid Insoluble | Iron (Fe) | Dry Burn Reduction |
|---|---|---|---|---|---|
| Proportion (%) | ≥96.0 | ≤2.0 | ≤0.1 | ≤0.05 | ≤0.5 |

**Table 8.** Physical properties of slaked lime.

| Test Item | Test Results | Specification Requirement |
|---|---|---|
| Apparent specific gravity | 2.325 | / |
| Specific surface area (m²/g) | 13.4644 ± 0.0468 | / |
| Exterior | No caking granules | / |

Waste brake pad powder: The waste brake pad powder used in the study comprised of waste brake pads purchased from a car repair shop that were crushed, grinded, and screened using a crusher. Its chemical composition and physical properties are shown Tables 9 and 10, respectively.

**Table 9.** Chemical composition of the brake pad powder.

| Element | SiO₂ | CaO | MgO | Fe₂O₃ | Al₂O₃ | SiO₃ | BaO | Others |
|---|---|---|---|---|---|---|---|---|
| Proportion (%) | 22.3 | 15.2 | 7.0 | 4.9 | 4.0 | 3.8 | 4.4 | 38.4 |

**Table 10.** Physical properties of the brake pad powder.

| Test Item | Test Results | Specification Requirement |
|---|---|---|
| Apparent specific gravity | 2.235 | / |
| Specific surface area (m²/g) | 1.7750 ± 0.0200 m²/g | / |
| Exterior | No caking granules | / |

### 2.1.3. Asphalt (Asphalt-Binder, Binder, Bitumen)

The base asphalt used was a Grade A No. 70 road petroleum asphalt, denoted as Pen70 in this study. As listed in Table 11, all its physical and rheological properties satisfactorily met the specification requirements of the Chinese standard JTG E20-2011 [32]. Note in this paper that the term "asphalt" has been used interchangeably to refer to "asphalt-binder", "binder", or "bitumen".

**Table 11.** Technical indices of the Pen70 asphalt.

| Test Items | | Units | Test Result | Specification Requirements |
|---|---|---|---|---|
| Penetration (25 °C, 5 s, 100 g) | | 0.1 mm | 63.4 | 60~80 |
| Penetration Index (PI) | | / | −0.88 | −1.5~+1.0 |
| Softening point (R&B) | | °C | 48.5 | ≥43 |
| 15 °C elongation | | cm | >100 | ≥100 |
| Wax content (distillation method) | | % | 1.0 | <2.2 |
| Flash point | | °C | 300 | ≥260 |
| Density (15 °C) | | g·cm$^{-3}$ | 1.036 | Measured |
| Solubility | | % | 99.9 | ≥99.5 |
| 60 °C dynamic viscosity | | Pa·s | 290 | ≥160 |
| Residual ductility (10 °C) | | cm | 7.8 | ≥6 |
| The remains after TFOT | Quality loss | % | 0.2 | ≤±0.8 |
| | Residual penetration ratio 25 °C | % | 68.0 | ≥61 |

Legend: R&B = Ring & Ball; TFOT = Thin Film Oven Test.

### 2.2. Filler Proportions and Preparation of the Asphalt Mortar

Limestone powder was used as the reference datum in a ratio of 1:1 with the asphalt to produce the reference or control asphalt mortar. Thereafter, the other filler materials were used to replace the limestone powder in an equal volume as shown in Table 12.

**Table 12.** Compositional proportions of the fillers in the asphalt mortars.

| Filler Material | Filler Quantity (g) | Volume (cm$^3$) | Base Asphalt (g) | Volume (cm$^3$) |
|---|---|---|---|---|
| Limestone powder | 150.00 | 56.84 | 150.00 | 144.79 |
| Waste brake pad powder | 127.04 | 56.84 | 150.00 | 144.79 |
| Cement | 174.61 | 56.84 | 150.00 | 144.79 |
| Slaked lime powder | 132.15 | 56.84 | 150.00 | 144.79 |

For the asphalt mortar preparation, the asphalt (Pen70) was oven heated at 135 °C for 4 h to liquefy it and allow for proper mixing with the fillers. Concurrently, the filler materials were also separately oven heated at 135 °C for 2 h to dry them. Thereafter, the asphalt and fillers were mixed in the proportions listed in Table 12 following the procedural process schematically illustrated in Figure 1.

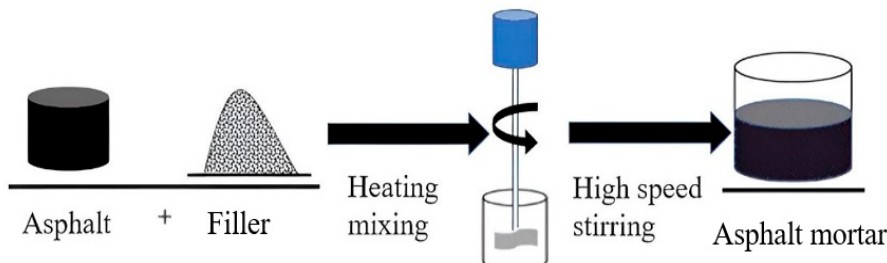

**Figure 1.** Preparation process of the asphalt mortar.

To prevent agglomeration, the weighed filler should be poured into the heated asphalt progressively in small amounts and consistently stirred at a constantly controlled temperature of 135 ± 5 °C. The stirring speed of the rotary agitator was 2000 ± 200 r/min. The asphalt and filler blend mixture were thereafter stirred for approximately 30 min to ensure a homogenous asphalt mortar matrix system. For each filler type, a minimum of three asphalt mortar sample replicates were prepared. Note also in this paper that the term "asphalt mortar" was used interchangeably with the terms "asphalt mastic" and "asphalt paste".

### 2.3. Mix-Design Proportions and Preparation of the Asphalt Mixture

The top surfacing layer of an asphalt pavement structure is considered the layer that is exposed to the harshest environmental conditions including rain, moisture intrusion, and water damage. For this reason, a typical dense graded AC-13 asphalt mixture, meeting the Chinese specification JTGF40-2004 [33] for the surfacing asphalt mixture layer, was used. The aggregate gradation, primarily limestone aggregates, is shown in Figure 2. In the asphalt mixture matrix, the asphalt mortar content was 4.58% by weight of the whole asphalt mixture. For each filler type, a minimum of three asphalt mixture samples were molded using the Superpave Gyratory Compactor (SGC) method and fabricated to 96 ± 1% and 93 ± 1% densities.

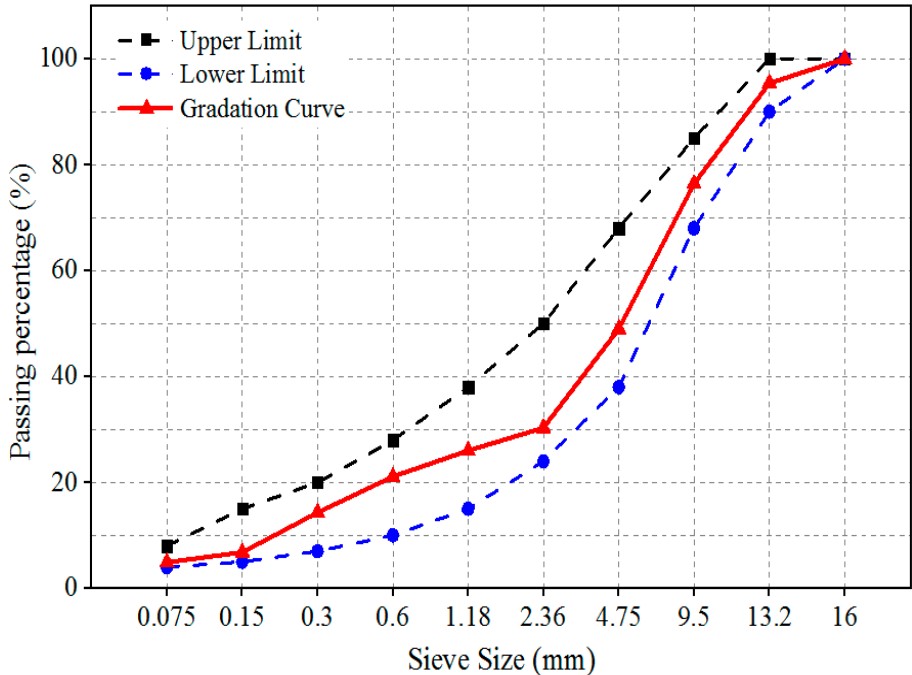

**Figure 2.** Aggregate (limestone) gradation curve for the AC-13 asphalt mixture.

### 2.4. Laboratory Test Methods

For assessing the combined effects of moisture and temperature, the asphalt mixture samples were comparatively tested after water ($H_2O$) saturation for 1 day and 3 days at 60 °C, 25 °C, and 0 °C. For each test type, material and temperature water-saturation condition, three replicate samples were tested. For each test type, material property and performance indicator, the generalized Equation (1) was blanketly used to quantitatively characterize the moisture sensitivity and water damage resistance of the asphalt mixtures, before and after water saturation:

$$A = \frac{P_0 - P}{P_0} \tag{1}$$

In Equation (1), $A$ is the performance loss rate due to moisture and water saturation effects; $P_0$ is the performance indicator corresponding to a specific test type and material property before water saturation; and $P$ is the performance indicator corresponding to a specific test type and material property after water saturation.

#### 2.4.1. Moisture Sensitivity Evaluation

Freeze-thaw splitting tests were used to evaluate the moisture sensitivity and water damage resistance of the asphalt mixtures with different fillers. The tests were conducted according to the Chinese specification JTG E20-2011 [32]. The test methods involved fabricating SGC samples with diameter of 101.6 mm and height of 63.5 mm at an air void level of 7 ± 1%. Thereafter, three replicate samples per filler type were subjected to

freeze-thaw splitting at 25 °C to determine their indirect tensile strength before and after moisture conditioning. From the indirect tensile strength results before and after moisture conditioning, the tensile strength ratio (TSR) was subsequently computed to quantify the moisture sensitivity and water damage resistance of the asphalt mixtures with different filler materials.

### 2.4.2. High-Temperature Stability Evaluation

Dynamic uniaxial compression tests [34] were used to evaluate the high-temperature performance of the asphalt mixtures with different filler materials. The UTM-100 universal testing machine was used for performing the test on standard SGC molded samples with a diameter of 100 mm and height of 100 mm that were compacted to 4% air voids. The test loading parameters comprised of 0.7 Mpa input stress, 0.1 s loading time, and 0.9 s resting time of the half sine wave at 60 °C [35,36]. The test was terminated when the sample's vertical displacement reached 10 mm. The number of repetitive load cycles and/or testing time to failure were considered as indicative of the asphalt mixture's high temperature stability and rutting resistance performance.

### 2.4.3. Low-Temperature Crack Resistance Evaluation

The Semicircular Bending (SCB) [37] test at −10 °C was used to evaluate the low-temperature cracking performance of the asphalt mixture with different filler materials. The test samples, 150 mm in diameter and 50 mm thick, were prepared using the SGC to 4% air voids. The SCB test was conducted using the UTM-100 machine (Figure 3) with the samples incubated and temperature conditioned for five hours at −10 °C prior to testing. The SCB loading rate was 50 mm/min until fracture failure. From the output data, namely the load/stress and displacement/strain, the fracture energy was computed to quantify the low-temperature crack resistance of the asphalt mixtures with different filler materials.

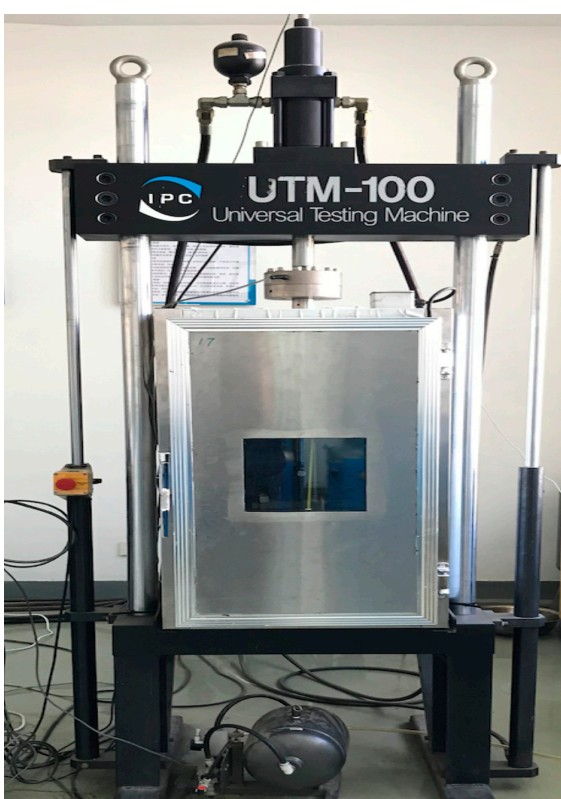

**Figure 3.** The UTM-100 machine setup.

2.4.4. Fatigue Resistance Performance Evaluation

The Repeated Semicircular Bending (R-SCB) [38] test was used to evaluate the fatigue performance of the asphalt mixtures with different fillers. SGC molded samples that were fabricated to 4% air voids with the same configuration and dimensions (i.e., 150 mm in diameter and 50 mm thick) as the SCB test were used. The R-SCB input load was determined from SCB testing at 25 °C as a fractional ratio of the SCB peak failure load [39,40]. In this study, the R-SCB test was conducted at five fractional stress ratios (the ratio of the actual applied load to the maximum SCB peak failure load) of the SCB peak load, namely 40%, 50%, 60%, 70%, and 80%. The number of repetitive load cycles to crack failure was used as an indicative measure of the asphalt mixture's fatigue resistance performance before and after moisture conditioning.

### 3. Laboratory Results and Discussions

*3.1. Moisture Sensitivity Analysis and Synthesis*

After curing the asphalt mixtures under dry and water saturated conditions for 1 day and 3 days at 0 °C, 25 °C, and 60 °C, IDT freeze-thaw splitting tests at 25 °C were conducted to measure and quantify the TSRs as shown in Figure 4. The corresponding TSR loss rates before and after water saturation are shown in Figure 5. In Figure 4, HL, BP, LP, and CP refers to hydrated lime, brake pads, cement pads, and limestone powder, respectively. Whereas 1 d/0 °C, 1 d/25 °C, 1 d/60 °C, 3 d/0 °C, 3 d/25 °C, and 3 d/60 °C in both Figures 4 and 5 refer to 1-day 0 °C, 1-day 25 °C, 1-day 60 °C, 3-day 0 °C, 3-day 25 °C, and 3-day 60 °C water saturation temperature conditions, respectively.

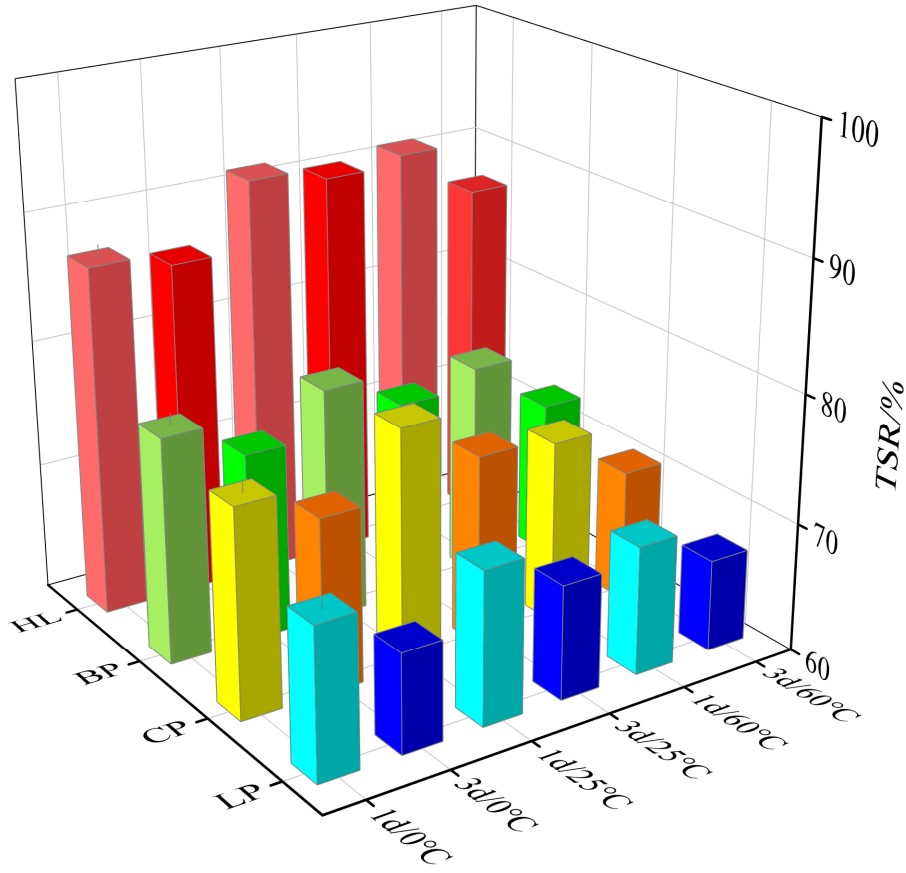

**Figure 4.** TSR test results of the asphalt mixtures.

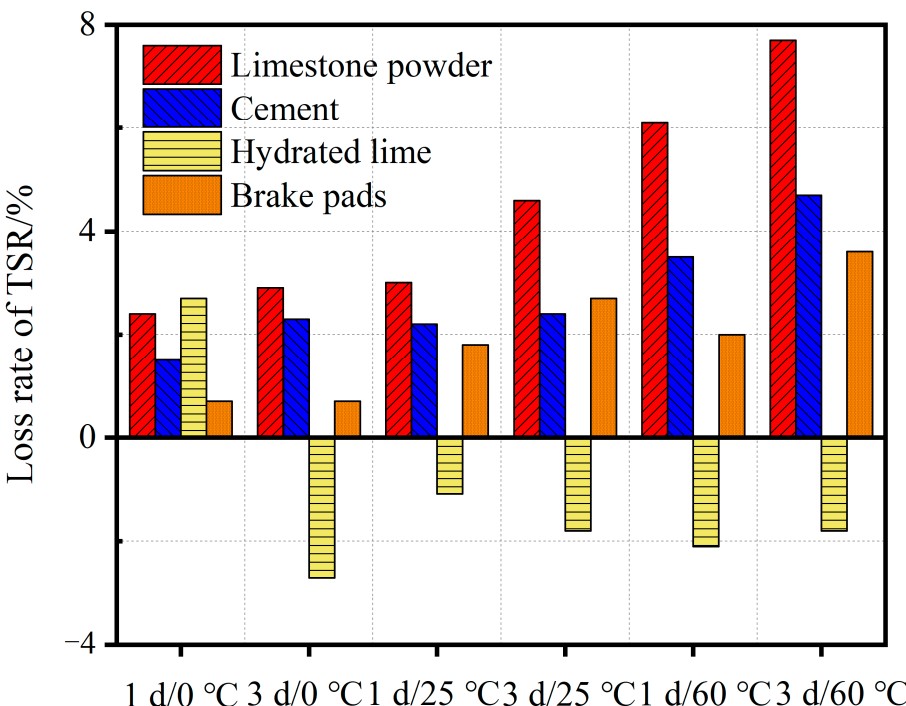

**Figure 5.** TSR loss rate of the asphalt mixtures after water saturation.

From Figure 4, the asphalt mixture with slaked lime (HL) exhibited the greatest moisture tolerance and water damage resistance, with the highest TSR values exceeding 80% after water saturation. If the 25 °C 1-day water saturation condition is considered for example, the TSR value of the asphalt mixture with slaked lime (HL) is 91.5%, which is 27.3%, 16.2%, and 17.0% higher than those of the asphalt mixtures with limestone mineral powder (LP), cement (CP), and brake pads (BP), respectively. As theoretically expected, water saturation, curing temperature, and time were all observed to have detrimental impacts on the TSR performance of the asphalt mixtures [41]. The TSR values of the asphalt mixtures with limestone mineral powder (LP), cement (CP), slaked (hydrated) lime (LP), and brake pads (BP), for example, decreased by approximately 4.2%, 4.8%, 1.1%, and 3.5%, respectively, when the 25 °C water saturation time was increased from 1 day to 3 days.

For illustrating the temperature's degradation effects, the TSR value of the asphalt mixture with limestone mineral powder (LP), for example, increased by 1.8% from 67.7% to 68.9% after the water saturation temperature was increased from 0 °C to 25 °C. At 60 °C, the corresponding TSR value was 67.1%, which is 0.8% lower than 0 °C and 2.6% lower than 25 °C. This phenomenon occurred partially because an increase of the temperature at 25 °C compared with 0 °C has a curing effect on the asphalt mixture. In this process, the curing effects of temperature on the asphalt mixture is greater than the damage impacts of the water to the asphalt mixture. However, with a further increase in temperature, for example, to 60 °C, the water caused more damage to the bonding interface between the asphalt and aggregates as well as the internal bonding properties of the asphalt mortar itself. Ultimately, the net damage effects of the water were greater than the curing effect of the temperature increment on the asphalt mixture—hence, an overall decay in the TSR value.

In Figure 5, the TSR loss rate of the asphalt mixture with slaked (hydrated) lime after water saturation is negative (except for 0 °C 1-day water saturation condition). That is, its TSR improved after water saturation treatment, which is an opposite response trend to the other filler materials. This negative TSR loss rate and increase in the TSR value of the asphalt mixture with slaked (hydrated) lime after water saturation could be partially attributed to the fact that the hydrated lime will react vigorously with the asphalt in the presence of water, forming a reinforced-like structure that greatly improves the splitting

resistance performance of the asphalt mixture [42–44]. The net result is an increase in the TSR value after moisture conditioning and a negative TSR loss rate. This also explains its (slaked lime) superior TSR performance over other filler materials.

From Figure 5, it can also be inferred that the asphalt mixture with slaked (hydrated) lime, with the least TRS loss rates, had superior water stability, moisture tolerance properties, and water damage resistance potential followed consecutively by brake pads, cement, and last, limestone mineral powder. In general, and as theoretically expected, moisture presence (water saturation), high curing temperature, and longer water saturation time were all observed to have detrimental impacts on the TSR loss rates of the asphalt mixtures. That is the longer the saturation time, the greater the TSR loss rate. Additionally, under the same water saturation time/conditions, the higher the water saturation temperature, the greater the TSR loss rate and the greater the water damage on the asphalt mixture. Overall, both Figures 4 and 5 indicated that slaked (hydrated) lime was the most moisture tolerant and water damage resistant filler—whereas the most moisture sensitivity and water damage susceptible filler was limestone mineral powder.

### 3.2. High-Temperature Stability Analysis and Synthesis

After curing at 0 °C, 25 °C, and 60 °C under dry and water saturated conditions for 1 day and 3 days, the number of repetitive load cycles to failures ($N_d$) for quantifying the high-temperature stability of the asphalt mixtures, as measured through dynamic uniaxial test at 60 °C, are shown in Figure 6 [45–47]. The corresponding loss rates in the number of failure load cycles after water saturation are also shown in Figure 7. In Figure 6, HL, BP, LP, and CP refers to hydrated lime, brake pads, cement pads, and limestone powder, respectively. Whereas 1 d/0 °C, 1 d/25 °C, 1 d/60 °C, 3 d/0 °C, 3 d/25 °C, and 3 d/60 °C in both Figures 6 and 7 refer to 1-day 0 °C, 1-day 25 °C, 1-day 60 °C, 3-day 0 °C, 3-day 25 °C, and 3-day 60 °C water saturation temperature conditions, respectively.

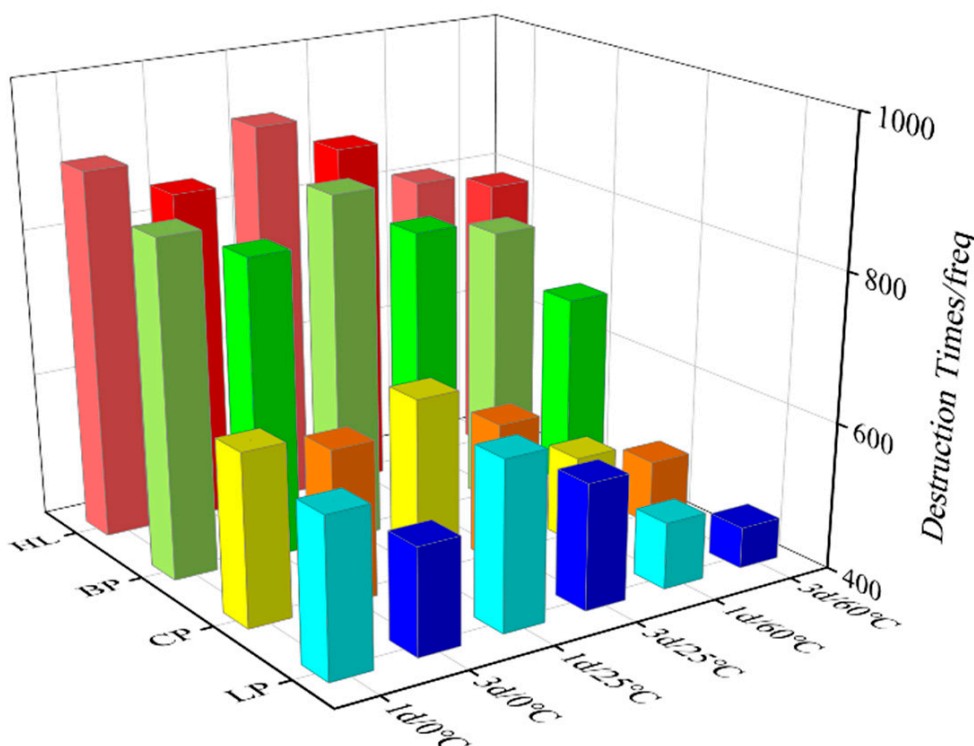

**Figure 6.** Dynamic uniaxial failure load cycles after water saturation.

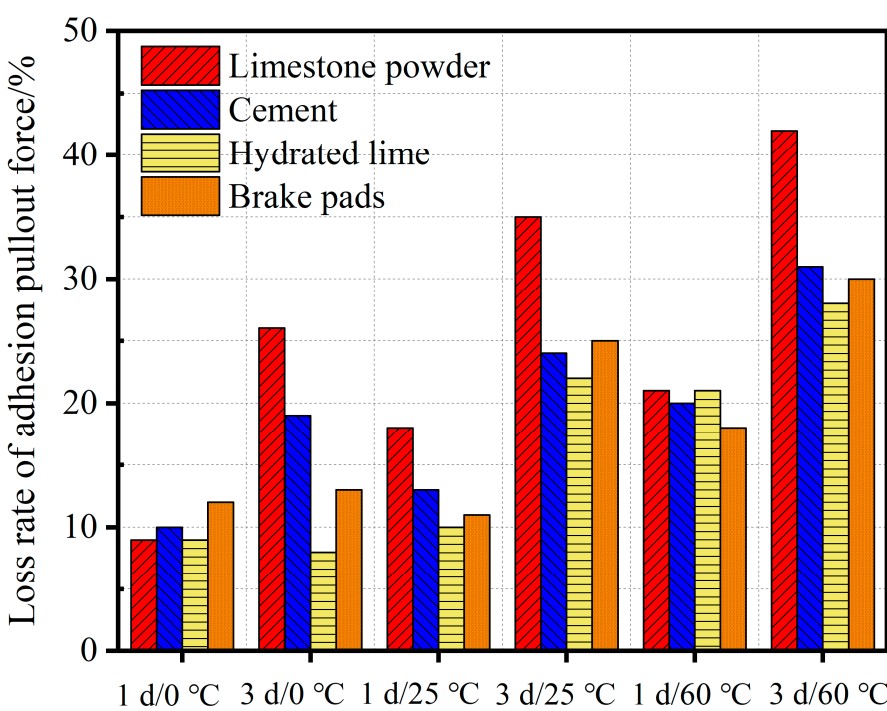

**Figure 7.** Loss rate in the failure load cycles after saturation.

From Figure 6, it is visually evident that the asphalt mixtures with slaked lime (HL) and brake pads (BP) performed comparably and significantly better than the asphalt mixtures with limestone mineral powder (LP) and cement (CP) fillers. Considering the 25 °C 1-day water saturation temperature, for example, the asphalt mixture with slaked (hydrated) lime sustained approximately 923 repetitive load cycles to failure which is approximately 46.3%, 41.6%, and 5.8% higher than the asphalt mixtures with limestone mineral powder (LP), cement (CP), and brake pad (BP) fillers, respectively. With an increase in the water saturation time, the number of repetitive load cycles decreased for all asphalt mixtures with slaked lime (HL) exhibiting the least reduction and the highest high-temperature stability potential. Taking the saturated water temperature of 25 °C as an example, the decline in the number of repetitive load cycles to failure for the asphalt mixtures with limestone mineral powder (LP), cement (CP), slaked lime (HL), and brake pad (BP) fillers were 9.4%, 9.0%, 5.1%, and 7.9%, respectively—ultimately indicating superior performance for slaked (hydrated) lime.

Likewise, both water saturation time and temperature had a negative impact on the high-temperature stability of the asphalt mixtures. However, whilst a slight performance improvement (i.e., increase in $N_d$) was noted for a temperature change from 0 °C to 25 °C under the short-term 1-day saturation period, a significant decay in the high-temperature stability of the asphalt mixtures was registered when the temperature was increased from 25 °C to 60 °C [48]. For example, the number of repetitive load cycles to failure of the asphalt mixture with limestone mineral powder increased by 2.6% from 615 to 631 cycles after the water saturation temperature was increased from 0 °C to 25 °C. When the water-saturation temperature was increased from 25 °C to 60 °C, however, the number of repetitive load cycles reduced by approximately 22.2% to 491 cycles.

In Figure 7, the asphalt mixture with slaked (hydrated) lime exhibited the best high-temperature stability with the lowest $N_d$ loss rates after water saturation followed consecutively by brake pads, cement, and limestone mineral powder. Ultimately, this means that the asphalt mixture with slaked (hydrated) lime filler had the best resistance to water damage and superior high-temperature stability performance. Under similar water saturation temperatures, the longer the water saturation time, the greater the loss rate and the greater the decay in the high-temperature stability of the asphalt mixture. For the same water

saturation period, the higher the water saturation temperature, the greater the $N_d$ loss rate and correspondingly, the greater the decay in the high-temperature stability of the asphalt mixture. Overall, both Figures 6 and 7 indicated the following rank order of superiority with respect to moisture tolerance and high-temperature stability: slaked (hydrated) lime > brake pads > cement > limestone mineral powder.

### 3.3. Low-Temperature Crack Resistance Analysis and Synthesis

After curing at 0 °C, 25 °C, and 60 °C under dry and water saturated conditions for 1 day and 3 days, the low-temperature cracking resistance of the asphalt mixtures was evaluated using the SCB test at −10 °C [39]. The SCB test results, with the low-temperature cracking resistance of the asphalt mixtures measured and quantified in terms of the fracture energy (FE), are shown in Figure 8. The corresponding FE loss rate after water saturation is shown in Figure 9. In Figure 8, HL, BP, LP, and CP refers to hydrated lime, brake pads, cement pads, and limestone powder, respectively. Whereas 1 d/0 °C, 1 d/25 °C, 1 d/60 °C, 3 d/0 °C, 3 d/25 °C, and 3 d/60 °C in both Figures 8 and 9 refer to 1-day 0 °C, 1-day 25 °C, 1-day 60 °C, 3-day 0 °C, 3-day 25 °C, and 3-day 60 °C water saturation temperature conditions, respectively.

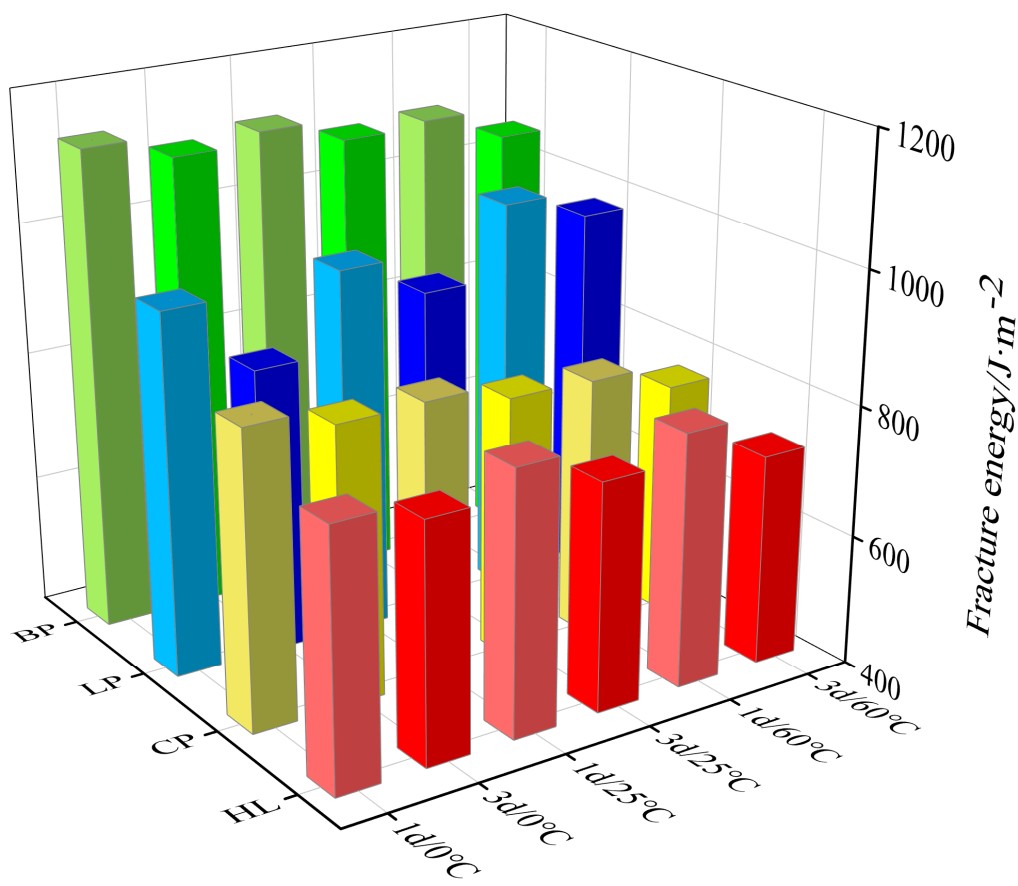

**Figure 8.** SCB fracture energy results of the asphalt mixtures.

From Figure 8, it is visually clear that the superiority rank order of low-temperature cracking resistance potential of the asphalt mixtures based on the quantitative magnitudes of the FE values is brake powder (BP), limestone mineral powder (LP), cement (CP), and slaked lime (HL). Considering to the 25 °C 1-day water saturation condition as an example, the FE of the asphalt mixture with brake pad (BP) filler is 1119 J/m². This is approximately 16.2%, 34.7%, and 39.0% higher than that of the asphalt mixture with limestone mineral powder (LP), cement (CP), and slaked (hydrated) lime (HL), respectively. Additionally, the figure also shows that longer water saturation periods generally consumed the FE with a

corresponding decay in the low-temperature cracking resistance for the asphalt mixtures. However, there were no definitive trends in the FE response behavior at high-temperature curing conditions. For example, the FE values of the asphalt mixtures with limestone mineral powder (LP), cement (CP), slaked (hydrated) lime (HL), and brake pad (BP) fillers decreased by 6.3%, 3.0%, 6.7%, and 3.0%, respectively, when the 25 °C water saturation time was increased from 1 day to 3 days. After 3 days of water saturation treatment, the FE of the asphalt mixture with limestone mineral powder (LP) filler increased by 7.2% from 842 J/m$^{-2}$ to 903 J/m$^{-2}$ when the water saturation temperature rose from 0 °C to 25 °C. At 60 °C, the FE rose to 972 J/m$^{-2}$, which is 15.4% higher than that of 0 °C and 7.6% higher than that of 25 °C. For the other filler materials, however, a progressive declining response trend was observed in the FE values when increasing the temperature from 0 °C through to 60 °C. Observing the 3-day water saturation condition for the asphalt mixture with slaked lime (HL), the FE dropped by approximately 1.8% from 765 J/m$^{-2}$ to 751 J/m$^{-2}$ when the water saturation temperature was increased from 0 °C to 25 °C. At 60 °C, the FE for slaked lime (HL) decayed to 723 J/m$^{-2}$, a decrease of approximately 5.5% compared to 0 °C and 3.7% compared to 25 °C.

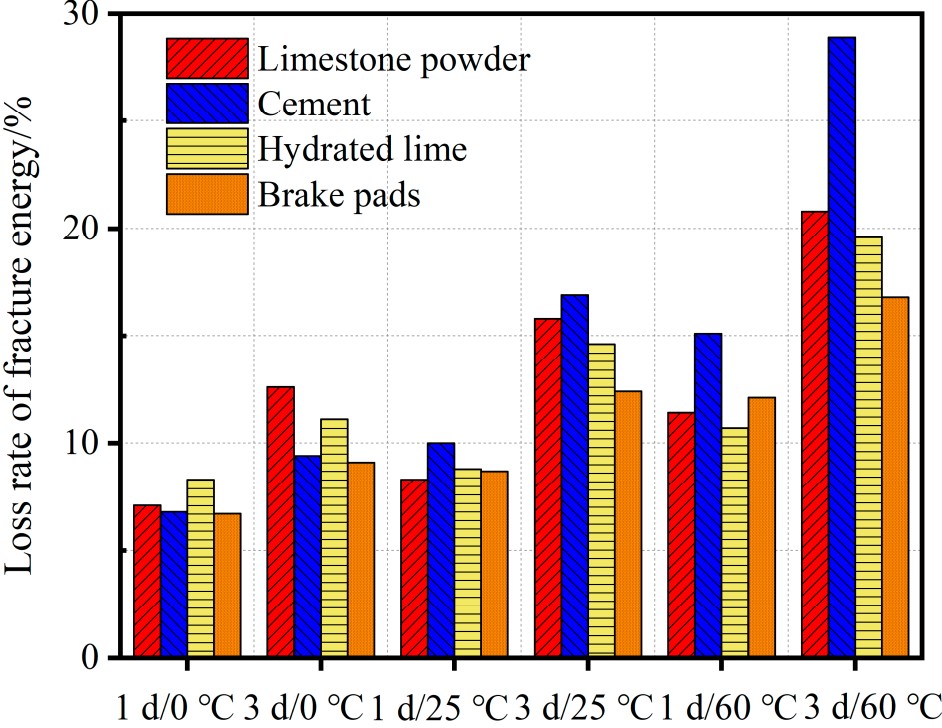

**Figure 9.** FE loss rates of the asphalt mixtures after water saturation.

In Figure 9, a decay in the low-temperature cracking resistance potential of the asphalt mixtures is evident as indicated by the increase in the FE loss rate after water saturation. Furthermore, the figure indicates that the FE loss rates are higher and more pronounced with increasing water saturation time and curing temperature. That is, the longer the water saturation time, the greater the FE loss rate. Likewise, the higher the water-saturation temperature, the higher the FE loss rate. Similarly, the greater the damaging impact of the water, the greater the decay in the low-temperature cracking resistance potential of the asphalt mixture.

In Figure 9, the asphalt mortar with brake pad filler exhibited the lowest FE loss rate, indicating superior resistance to water damage and low-temperature cracking. At high-temperature water saturation conditions, the asphalt mixture with cement performed the poorest with the highest FE loss rates. At the 60 °C 3-day water saturation condition, its FE loss rate was as high as 29.3%. These results suggests that the low-temperature cracking

resistance of an asphalt mixture will likely be more detrimentally affected by water under prolonged rainy summer seasons. Based on Figures 8 and 9, the generalized rank order of filler superiority with respect to low-temperature cracking resistance potential is brake pads > slaked (hydrated) lime > limestone mineral powder > cement.

### 3.4. Fatigue Resistance Analysis and Synthesis

After curing at 0 °C, 25 °C, and 60 °C for 1 day and 3 days under dry and water-saturated conditions, the fatigue resistance of the asphalt mixtures with different fillers was measured using the R-SCB test at 25 °C [49,50]. As measured in terms of the fatigue life (i.e., the number of repetitive tensile load cycles to crack failure), Figure 10 shows a logarithmic-linear decay in the fatigue resistance with an increase in the stress ratio (i.e., 0.40 versus 0.80); the stress ratio is the quantitative ratio of the actual applied load to the maximum SCB peak failure load. Likewise, a generalized declining response trend in the fatigue life was also noted when both the water saturation time (i.e., 1 day versus 3 days) and temperature (i.e., 0 °C versus 60 °C) were increased.

When comparing the filler materials, it is visually evident in Figure 10a,b that the asphalt mixture with limestone mineral powder performed the poorest with the least fatigue life, particularly under water saturated conditions at high stress levels. At higher stress levels, the fatigue life decreased significantly with an increase in the water saturation temperature, with the lowest fatigue recorded at 80% (i.e., 0.80) stress level for the 60 °C 3-day water saturation conditions. By contrast, the decay in fatigue life at low-stress levels (i.e., 0.40 to 0.50), low-temperatures (0 °C to 25 °C), and shorter water saturation periods (i.e., 1-day) were not as significantly detrimental as at higher stress levels (i.e., 0.80), high-temperature (60 °C), and longer water saturation periods (i.e., 3-days). In general, the fatigue lives in Figure 10 indicated an undesirable degradation in the asphalt mixtures' fatigue resistance under water saturation, longer curing time (3-days), high curing temperature (60 °C), and high SCB stress (0.80) test conditions. In terms of the filler materials, slaked (hydrated) lime exhibited the best cracking resistance with the highest fatigue life consecutively followed by brake pads and cement, and lastly, limestone mineral powder (i.e., poorest performer).

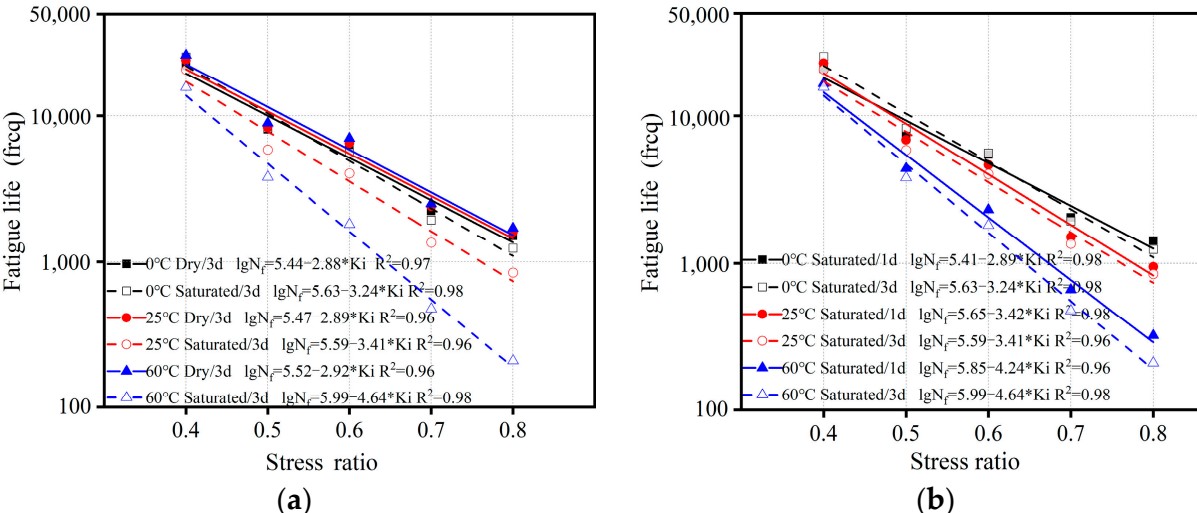

(**a**)                    (**b**)

**Figure 10.** *Cont.*

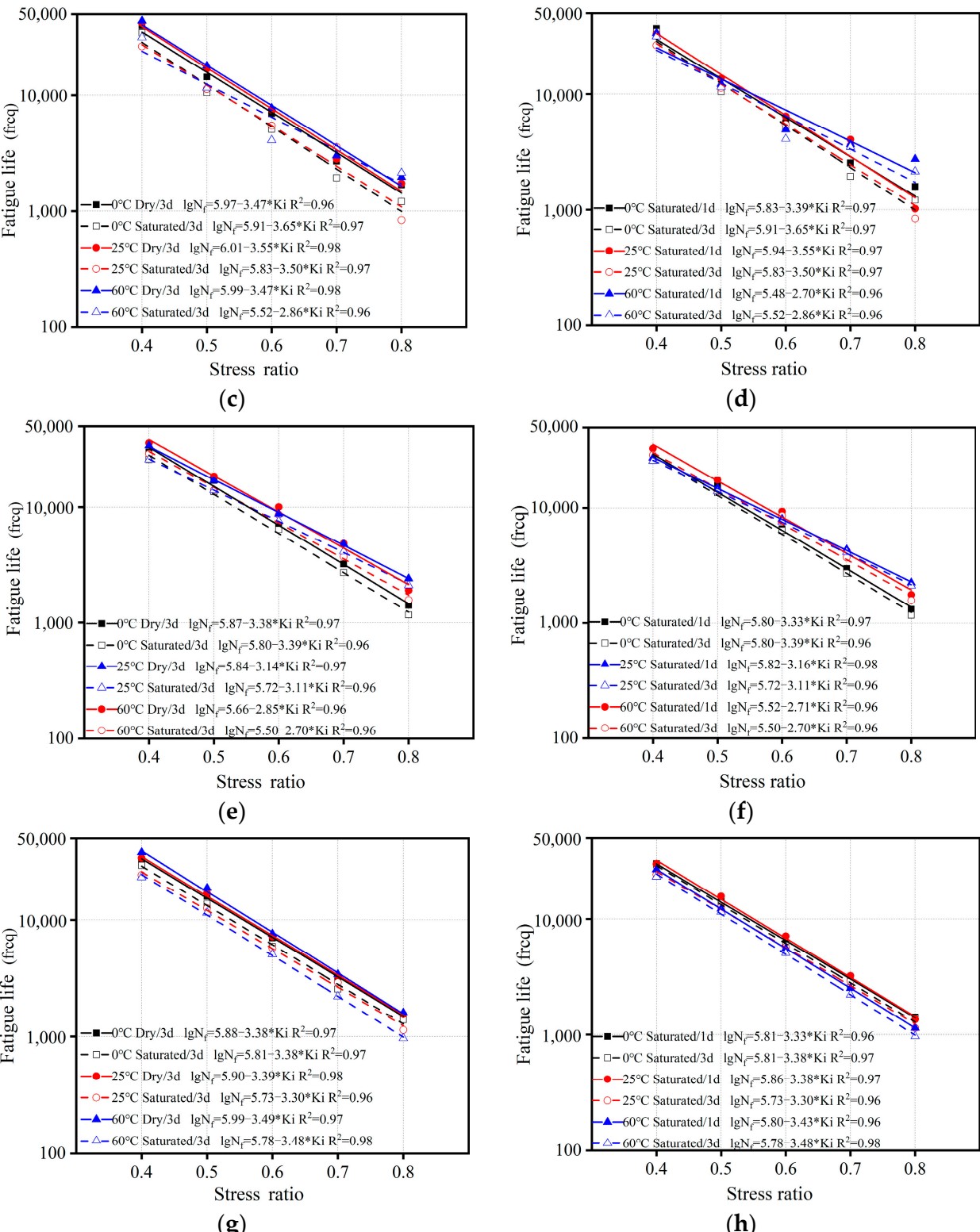

**Figure 10.** Fatigue life of the asphalt mixtures before and after water saturation (stress ratio = ratio of actual applied load to maximum SCB peak failure load). (**a**) Mineral powder—dry and $H_2O$ saturation, (**b**) Mineral powder—after $H_2O$ saturation, (**c**) Cement—dry and $H_2O$ saturation, (**d**) Cement—after $H_2O$ saturatioI (**e**) Slaked lime—dry and $H_2O$ saturation, (**f**) Slaked lime—after $H_2O$ saturation, (**g**) Brake pads—dry and $H_2O$ saturation, (**h**) Brake pads—after $H_2O$ saturation. (* is the multiplication sign in mathematical symbols).

## 4. Conclusions

In this laboratory study, a Pen70 base asphalt and four different filler materials, namely limestone mineral powder, cement, slaked (hydrated) lime, and brake pads were used to prepare asphalt mixtures. The relevant mechanical properties were comparatively evaluated, namely before and after water saturation conditions for 1~3 days at 0~60 °C curing temperatures. In the study, 25 °C 1-day curing time served as the control condition for the laboratory experimentation. From the study results and findings, the following conclusions were drawn:

(1) As theoretically expected, moisture conditioning and water saturation yielded negative impacts on the mechanical properties of the asphalt mixtures, with a general decay in the tensile strength (i.e., TSR), high-temperature stability (i.e., $N_d$), cracking resistance (i.e., FE), and fatigue life (i.e., $N_f$). However, the addition of mineral fillers, such as slaked (hydrated) lime, tended to enhance the moisture tolerance and water damage resistance of the asphalt mixtures.

(2) After water saturation treatment, the performance of the asphalt mixture decreased due to water damage, in particular the low-temperature cracking properties of the asphalt mixture were significantly degraded. Under high-temperature water saturation conditions, the fracture energy (FE) loss rate was more than 20%, with the cement asphalt mixture incurring the greatest FE degradation of approximately 29.3%.

(3) The degree of water damage to the asphalt mixtures was found to be significantly affected by temperature and water saturation time. Both high temperatures and prolonged water saturation periods tended to detrimentally aggravate the impacts of moisture damage on the asphalt mixtures with the latter being more impactful.

(4) In terms of filler material comparisons, the study results indicated the following rank order of superiority with respected to moisture tolerance, mitigating water damage, optimum mechanical properties, and good performance for the asphalt mixtures: slaked (hydrated) lime > brake pads > cement > limestone mineral powder. That is, for the materials evaluated and the laboratory test conditions considered, limestone mineral powder was found to be the most moisture-sensitive filler material, whilst slaked (hydrated) lime was observed to be the most moisture-tolerant and water-damage resistant filler material.

Overall, the study findings were plausible and indicated superior laboratory performance for the slaked (hydrated) lime filler (consecutively followed by brake pads, cement, and limestone mineral powder). For future follow-up studies, however, additional laboratory testing with different base asphalts, filler contents, material types, asphalt mortars, mix-designs, and asphalt mixtures along with field validation is recommended to further supplement and substantiate the results/findings reported herein. Nonetheless, the study valuably contributes to the literature enrichment on the use of different filler materials for modifying Pen70 asphalt to enhance the moisture tolerance, water damage resistance, and mechanical properties of asphalt mixtures.

**Author Contributions:** Conceptualization, T.B.; supervision, X.Z.; writing—original draft, Y.L.; data curation, F.W.; investigation, C.C.; writing—review and editing, C.P.; software, L.F.; methodology, L.F.W.; investigation, W.L.; resources, X.W. All authors have read and agreed to the published version of the manuscript.

**Funding:** This work is partially supported by National Natural Science Foundation of China (Grant No. 52108415, No. 52108425, and No. 51803157), Natural Science Foundation of Hubei Province (Grant No. 2020CFB567), the Enterprise Technology Innovation Project of Shandong Province (202160101791), and the Science and Technology Project of Shandong Hi-Speed Maintenance Group Co., Ltd. (2021-05).

**Institutional Review Board Statement:** Not applicable.

**Informed Consent Statement:** Not applicable.

**Data Availability Statement:** All of the data that support the findings of this study are available from the corresponding author upon reasonable request.

**Acknowledgments:** We express our sincere gratitude to the experts, teachers and students who have provided help for this article.

**Conflicts of Interest:** The authors declare no conflict of interest.

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
