# Peer review of "Assessing the Effects of Different Fillers and Moisture on Asphalt Mixtures’ Mechanical Properties and Performance"

_coatings, doi:10.3390/coatings13020288_

Round 1
Reviewer 1 Report
1: Improve the literature review part by addition of recent published papers.
Done – As directed by the Reviewer, the literature review section has been enhanced with additional recent publications in the revised manuscript. These additions include the following 10 recent publications:
Soenen, H., Vansteenkiste, S., & Kara De Maeijer, P. (2020). Fundamental approaches to predict moisture damage in asphalt mixtures: State-of-the-art review. Infrastructures, 5(2), 20.
Omar, H. A., Yusoff, N. I. M., Mubaraki, M., & Ceylan, H. (2020). Effects of moisture damage on asphalt mixtures. Journal of Traffic and Transportation Engineering (English Edition), 7(5), 600-628.
Chen, Y., Xu, S., Tebaldi, G., & Romeo, E. (2022). Role of mineral filler in asphalt mixture. Road materials and pavement design, 23(2), 247-286.
Das, A. K., & Singh, D. (2021). Evaluation of fatigue performance of asphalt mastics composed of nano hydrated lime filler. Construction and Building Materials, 269, 121322.
Gedik, A. (2021). An exploration into the utilization of recycled waste glass as a surrogate powder to crushed stone dust in asphalt pavement construction. Construction and Building Materials, 300, 123980.
Li, Y., Hu, X., Zhao, Y., Zhu, G., Wang, N., Pan, P., ... & Sun, Y. (2022). Performance evaluation of asphalt mixture using brake pad waste as aggregate. Case Studies in Construction Materials, 17, e01639.
Razavi, S. H., & Kavussi, A. (2020). The role of nanomaterials in reducing moisture damage of asphalt mixes. Construction and Building Materials, 239, 117827.
Preti, F., Accardo, C., Gouveia, B. C. S., Romeo, E., & Tebaldi, G. (2021). Influence of high-surface-area hydrated lime on cracking performance of open-graded asphalt mixtures. Road Materials and Pavement Design, 22(11), 2654-2660.
Awed, A. M., Tarbay, E. W., El-Badawy, S. M., & Azam, A. M. (2022). Performance characteristics of asphalt mixtures with industrial waste/by-product materials as mineral fillers under static and cyclic loading. Road Materials and Pavement Design, 23(2), 335-357.
Dimulescu, C., & Burlacu, A. (2021). Industrial Waste Materials as Alternative Fillers in Asphalt Mixtures. Sustainability, 13(14), 8068.
2. Revise figure 1.
Done – This arose when converting from Word to PDF. It should be good in the revised manuscript now.
3. Revise title 3. Page 8 line 233.
Done – The title has been corrected accordingly.
4. Improve the discussion of results.
Done – The discussions were revised and improved accordingly.
5. Revise the conclusion part.
Done – The discussions were revised and improved accordingly.
Reviewer 2 Report
Mandatory requirements:
1: LNSA (Line Number Scientific Article) 12-24….Abstract… I would allow myself to take the following position regarding the abstract. In general, I expect from a high-quality scientific abstract a brief scientific summary of the solved problem, an explicit determination of scientific goals and corresponding methodology. In the abstract, a global view is completely absent, as well as a clear justification of the chosen asphalt modification methods. The sentence "Overall, the study findings indicated a degradation in the mechanical properties and performance of the asphalt mixtures after moisture conditioning and saturation in water, particularly the low-temperature cracking resistance characteristics" I recommend to restyle. Shouldn't the reason for the modification be the improvement of at least some properties of the resulting asphalt mixture?
Done – The abstract has been revised as directed.
2:LNSA 106…Table 1. Physical and technical properties of the coarse aggregates.... Error! Reference source not found...je potrebné nahradiÅ¥ správnou odvolávkou, platí aj pre Tables 2, 4-6, 8, 10 and 11.
Done – The references have been corrected as directed. This arose when converting from Word to PDF. It should now be good in the revised manuscript.
3: LNSA 159... Figure 1. Preparation process of the asphalt mortar... the picture needs to be reworked, only part of the texts can be seen.
Done – The references have been corrected as directed. This arose when converting from Word to PDF. It should now be good in the revised manuscript.
4: LNSA 187... Equation (1)...in the scientific literature, the standard numbering of equations is only (1).
Done – The format has been corrected by deleting the word “Equation”.
5: LNSA 202... Error! Reference source not found....it is necessary to indicate the correct reference.
Done – Corrections have been implemented as directed.
6: LNSA 221-222…materi-als…incorrectly split word.
Done – Corrections have been implemented as directed. This was a typo.
7: LNSA 233...3. labORATORY results and discussions...incorrect capitalization.
Done – Corrections have been implemented as directed. This was a typo.
8: LNSA 396… Based on Figures 25 and 26…wrong figure numbers.
Done – Corrections have been made as directed. This was a typo. Additionally, a figure for the UTM-100 as been added as Figure 3 and the figure numbers have been all adjusted accordingly throughout the revised manuscript.
9: LNSA 407... Figure 9. Fatigue life of the asphalt mixtures before and after water saturation...it is necessary to add physical units to the individual axes in the graphs in figure 1, or at least specify them in the title of the figure.
Done – Corrections have been made as directed. Stress ratio has now been explicitly defined in Sections 2.4.4 and 3.4 as well as under Figure 10 of the revised manuscript.
10: LNSA 475-552…References…need to unify the use or non-use of a dot at the end of the relevant reference.
Done – Corrections have been made as directed to consistently unify the references.
Facultative recommendations:
11: LNSA 2-3... Assessing the Effects of Different Fillers and Moisture on Asphalt Mixtures’ Mechanical Properties and Performance... I would like to recommend to the authors a slight modification of the title of reviewed scientific article. Personally, I would use, for example, the following title "Assessing of different fillers and their moistures on mechanical properties and performance of asphalt mixtures" (please take it only as an inspiration).
Thanks very much for the Reviewer’s insightful suggestion. The paper was focused on both the filler type and moisture effects assessment. Thus, we are of the opinion to leave the original title as it is.
12: LNSA 28-39… Highlights… I recommend the authors consider omitting this part of the article. In the presented form, the texts are substantially duplicated with the abstract and, from my point of view, are redundant in the given form. So far, I have not met "Highligts" in the review proceedings for the publishing house MDPI, and I processed more than 55 reviews during the year 2022. If the authors decide to keep them, I recommend their revision and removal of explicit duplications with the abstract.
Done – Highlights have been deleted as directed by the reviewer.
13: LNSA 119... Table 3. Chemical composition of limestone... why is the SiO3 content specifically stated at the level of 0.1% and 41.7% is only characterized as Others?
Thanks very much for the reviewer’s observations. SiO3 has been deleted and included in others. The “Others (41.8%)” mean the elemental weight proportion less than 0.1% and the loss on ignition, where the X-Ray Fluorescence analysis was used.
14: LNSA 140... Apparent relative density... for readers not directly working in the subject research area, it would be appropriate to briefly explain this material characteristic.
Thanks for the insightful observation. This was a typo and has since been corrected to “apparent specific gravity” in the revised manuscript.
15: LNSA 205-206... The UTM-100 universal testing machine... I recommend that you consider adding a photo of the device, or at least explain the UTM abbreviation.
Done – A photo of the UTM-100 universal testing machine has been added as Figure 3 in the revised manuscript.
16: LNSA 424-469... 4. Summary and Conclusions ... I personally prefer to state the title of the chapter only in the form Conclusions or discussion and conclusions... A very good abstracting of the research findings has the potential for separate processing of chapters 4. Discussions and 5. Conclusions. In the conclusions, I recommend implementing a comparison of the author's research results with the most important works of foreign authors.
Thanks for the insightful observations. Both the discussions and synthesis aspects of Section 3 as well as the conclusions in Section 4 were revisited, revised, and enhanced accordingly.
Round 2
Reviewer 2 Report
Based on the incorporation of the changes recommended by me, I allow myself to rate the assessed second version of the scientific paper as follows. Reviewed contribution: "Assessing the Effects of Different Fillers and Moisture on Asphalt Mixtures’ Mechanical Properties and Performance", overall I rate it as excellent. Based on my experience in the assessed issue and subsequent deepening of my knowledge, I am pleased that the submitted 2nd version of the article meets all my essential requirements for a quality scientific article. I am fully satisfied with the implementation of the required changes and recommendations and I hope that I have contributed a little to improving the quality of the assessed scientific contribution. In conclusion, I would like to sincerely congratulate the authors on an excellent scientific article and thank the publisher for the opportunity to expand my scientific knowledge in the scientific assesment area of various environmental progressive filler materials for modifying asphalt to enhance in order to mechanical properties of asphalt mixtures.